# Signal Transduction Pathway Activity in High-Grade, Serous Ovarian Carcinoma Reveals a More Favorable Prognosis in Tumors with Low PI3K and High NF-κB Pathway Activity: A Novel Approach to a Long-Standing Enigma

**DOI:** 10.3390/cancers12092660

**Published:** 2020-09-18

**Authors:** Laura van Lieshout, Anja van de Stolpe, Phyllis van der Ploeg, David Bowtell, Joanne de Hullu, Jurgen Piek

**Affiliations:** 1Department of Obstetrics and Gynecology, Catharina Cancer Institute, Catharina Hospital, 5602ZA Eindhoven, The Netherlands; phyllis.vd.ploeg@catharinaziekenhuis.nl (P.v.d.P.); jurgen.piek@catharinaziekenhuis.nl (J.P.); 2Radboud Institute for Health Sciences, Department of Obstetrics and Gynecology, Radboud University Medical Center, 6500HB Nijmegen, The Netherlands; Joanne.deHullu@radboudumc.nl; 3Precision Diagnostics, Philips Research, 5656AE Eindhoven, The Netherlands; anja.van.de.stolpe@philips.com; 4Peter MacCallum Cancer Centre, 305 Grattan Street, Melbourne 3000, Australia; nadia.traficante@petermac.org; 5Centre for Cancer Research, The Westmead Institute for Medical Research, Sydney 2145, Australia

**Keywords:** high-grade serous ovarian cancer, epithelial ovarian cancer, signal transduction pathways, progression-free survival, cluster analysis

## Abstract

**Simple Summary:**

All cells have a complex internal network of ‘communication chains’ called signal transduction pathways (STPs). Through interaction of different proteins in STPs, they are partly responsible for the behavior of a cell. In our study, we investigated the activity of eight STPs in datasets with genetic information on 140 cancer samples. These samples were derived from the most common subtype of ovarian cancer: high grade serous ovarian carcinoma (HGSC). With a novel method, we determined which STPs were active and discerned two groups based on activity of the phosphoinositide 3-kinase (PI3K) and nuclear factor-kappa B (NF-kB) pathways. The group with low PI3K and high NF-kB activity had a better progression free and overall survival compared to the group with high PI3K and low NF-kB activity. This difference may indicate that the ‘better prognosis group’ had a more active immune system or that the cells divided at a slower rate.

**Abstract:**

We investigated signal transduction pathway (STP) activity in high-grade serous ovarian carcinoma (HGSC) in relation to progression-free survival (PFS) and overall survival (OS). We made use of signal transduction pathway activity analysis (STA analysis), a novel method to quantify functional STP activity. Activity of the following pathways was measured: androgen receptor (AR), estrogen receptor (ER), phosphoinositide 3-kinase (PI3K), Hedgehog (Hh), Notch, nuclear factor-kappa B (NF-κB), transforming growth factor beta (TGF-β), and Wnt. We selected HGSC samples from publicly available datasets of ovarian cancer tissue, and used repeated *k*-means clustering to identify pathway activity clusters. PFS and OS of the clusters were analyzed. We used a subset of publicly available dataset GSE9891 (*n* = 140), where repeated *k*-means clustering based on PI3K and NF-κB pathway activity in HGSC samples resulted in two stable clusters. The cluster with low PI3K and high NF-κB pathway activity (*n* = 72) had a more favorable prognosis for both PFS (*p* = 0.004) and OS (*p* = 0.001) compared to the high-PI3K and low-NF-κB pathway activity cluster (*n* = 68). The low PI3K and high NF-κB pathway activity of the favorable prognosis cluster may indicate a more active immune response, while the high PI3K and low NF-κB pathway activity of the unfavorable prognosis cluster may indicate high cell division.

## 1. Introduction

In 2018, approximately 300,000 cases of ovarian cancer were diagnosed worldwide [1]. Approximately 60% of these cancers are high-grade serous carcinomas (HGSCs). Despite aggressive treatment with a combination of surgery and chemotherapy, HGSC remains the most lethal gynaecological malignancy, with an overall five-year survival rate of only 30% for advanced stage HGSC [2]. Remarkably, both progression-free survival (PFS) and overall survival (OS) vary greatly among women diagnosed with advanced stage HGSC, despite histological resemblance and similar treatment. A subset of women succumbs to the disease within months, while other patients remain in complete remission for over a decade. As this variety cannot be explained by classic clinicopathological factors, the question arises whether other tumor characteristics can account for the variety in survival, such as differences in the underlying tumor driving mechanisms.

Cellular processes are controlled by signal transduction pathways (STPs), which govern cell division, migration, differentiation, and up to a certain degree, cell metabolism [3,4]. This process is tightly controlled, as aberrant STPs can result in uncontrolled and unlimited cell growth, and therefore be the driving force of tumor growth and metastasis [5]. Most tumor driving gene mutations known to date are part of one or more of 12 STPs [6]. Although much remains to be elucidated with respect to cancer genomics, treatment strategies are increasingly tailored to individual patients and their tumor characteristics. For these therapies to be effective, it is essential to target the right STP. Unfortunately, identification of the tumor-driving pathway has proven to be a challenge, which is illustrated by the fact that response rates to targeted therapies against STPs are often lower than expected [7,8]. Activity of STPs can be determined based on (over-)expression of key proteins of signaling pathways in combination with analysis of the activity state, or on the genotype of the cancer cell through identification of specific gene alterations. The problem with the latter approach is that it is based solely on the genotype of the cancer cell, rather than the functional phenotype, while evidence is accumulating that the phenotype is affected by other factors, such as the tumor micro-environment [7]. In 2010, Verhaegh et al. developed an approach to assess the cancer phenotype, using knowledge-based Bayesian network models to interpret a selection of the cancer transcriptome: signal transduction pathway activation analysis (STA analysis). It estimates functional pathway activity based on mRNA levels of target genes of the major oncogenic STPs [9]. Currently, eight major pathways can be analyzed by means of this model; estrogen receptor (ER), androgen receptor (AR), phosphoinositide 3-kinase (PI3K, measured indirectly as, in the absence of cellular oxidative stress, it is inversely proportional to forkhead box O (FOXO) activity), hedgehog (HH), Notch, transforming growth factor beta (TGF-β), nuclear factor-kappa B (NF-κB), and canonical Wnt [7,10,11,12,13]. In breast and colon cancer, STA analysis has already proven to be of value, as it accomplishes high accuracy in the determination of the aberrant STP [7]. In women with breast cancer, STA-analysis resulted in a more accurate prediction of response to neoadjuvant hormonal therapy when compared to traditional immunohistochemical ER staining [14]. Furthermore, STA analysis was able to identify a subgroup of patients with a more favorable prognosis among women diagnosed with ER-positive breast cancer treated with tamoxifen, based on actual ER pathway activity [7]. Similar findings are reported in endometrial cancer [15]. These results indicate that STA analysis may be a superior approach to assess STP activity and select patients for individualized therapies.

As classic clinicopathological factors cannot account for the variation in survival of women with HGSC, it may be the result of previously unnoticed differences in the underlying tumor driving mechanisms. We aim to investigate STP activity in publicly accessible datasets of ovarian cancer to assess possible differences in functional phenotype as an explanation for differences in PFS and OS of patients with HGSC.

## 2. Results

### 2.1. Datasets and Patient Characteristics

We searched the Gene Expression Omnibus (GEO) via https://www.ncbi.nlm.nih.gov/geo/ [16] for datasets containing Affymetrix data of HGSC with clinical annotations [16]. Our search resulted in two eligible datasets, GSE9891 and GSE32062, which were then subject to quality control. The data was subsequently analysed to provide pathway activity scores, as described in Methods Section 4.2. Activity scores for the AR, ER, PI3K, HH, NF-κB, Notch, TGF-β, and Wnt pathways were provided on a scale from 0 to 100.

#### 2.1.1. GSE9891

GSE9891 was used in a publication of Tothill et al. (data accessible at NCBI GEO database, accession GSE9891) [17]. The publication provides a detailed description of the sample processing, gene expression profiling, and molecular subtyping based on gene expression profiling [17]. In addition to the clinical annotations, we were able to obtain additional survival data as was known in October 2019 from the Australian Ovarian Cancer Study (AOCS) (including data from the Westmead GynBiobank), who provided the samples for this dataset (see Methods Section 4.2).

The dataset contained 285 tumor samples in total. Twenty-one samples were excluded from our analysis, as 18 samples were collected after the start of neo-adjuvant chemotherapy, and for three samples the timing of sampling was unknown (see Methods Section 4.2). The 264 samples collected prior to start of chemotherapy were suitable for analysis. A majority of the samples were serous carcinomas (*n* = 227), but the dataset also included samples of women diagnosed with endometrioid carcinoma (*n* = 18), serous borderline tumors (*n* = 18), and an adenocarcinoma not otherwise specified (NOS; *n* = 1). All patients were treated with debulking surgery. A total of 227 received subsequent chemotherapeutic treatment. Thirty-seven (37) patients received no additional chemotherapeutic treatment, most of which were patients diagnosed with a borderline tumor (*n* = 18).

For three samples, there was evidence of cellular oxidative stress (see Methods Section 4.1). All three samples were of serous carcinomas: one was grade 1, one grade 2, and one grade 3. The three individual samples are clearly marked in the results section.

In their paper, Tothill et al. describe six distinct molecular clusters within their dataset; the allocated cluster number was included in the dataset [17]. Figure 1 shows a flow diagram of included samples and the ways they can be grouped, according to classic histopathology or according to the clusters as formed by Tothill et al. [17]. An overview of patient and tumor characteristics of the included samples is provided in Table 1.

#### 2.1.2. GSE32062

GSE32062 was used in a publication of Yoshihara et al. (data accessible at NCBI GEO database, accession GSE32062). A detailed description of the sample processing and gene expression profiling can be found in the paper Yoshihara et al. [18]. The original dataset for this study contained 260 samples. However, due to the platform used for gene expression profiling, a subgroup of ten samples was suitable for STA analysis. All samples were of women diagnosed with HGSC. Age at time of diagnosis was unknown. Four women were diagnosed with FIGO [19] stage IV and six with FIGO stage IIIC. All women had debulking surgery and were treated with chemotherapy. Seven women had a recurrence and died, two women had a recurrence but were alive at the end of follow-up, and one woman remained free of disease for the duration of follow-up.

### 2.2. Clustering of HGSC Samples

#### 2.2.1. GSE9891

Preliminary data of a pilot study of van de Wiel et al. suggested that a combination of low PI3K and high NF-κB pathway activity might be associated with prolonged survival in women with HGSC [20]. Therefore, we selected primary serous tumor samples that were classified as grade 3 for clustering analyses (*n* = 140). Allocation to one of two clusters was based on PI3K (median = 55, range: 27–72) and NF-κB (median = 52, range: 17–83) pathway activity scores. Repeated *k*-means clustering resulted in two stable clusters: cluster A (*n* = 68) with a higher PI3K (median = 57, range: 45–72) and lower NF-κB pathway activity (median = 41, range: 17–52) compared to cluster B (*n* = 72) with lower PI3K (median = 52, range: 27–68) and higher NF-κB pathway activity (median = 62, range: 51–83). The initial survival analysis with publicly available survival data revealed a poorer prognosis for cluster A for both PFS (*p* = 0.018) and OS (*p* = 0.012, Figure 2a). A repeat survival analysis with additional follow-up data provided by the AOCS resulted in an increase in the difference in PFS (*p* = 0.004) and OS (*p* = 0.001), as shown in Figure 2b.

In only one of the primary HGSC samples included in the clustering analysis, the combination of high FOXO activity with high SOD2 mRNA expression was found, suggestive of cellular oxidative stress. As this influences interpretation with respect to associated PI3K pathway activity, we performed a post-hoc sensitivity analysis after exclusion of this sample (*n* = 139). Removal of the sample did not affect the difference in DFS (*p* = 0.006) or OS (*p* = 0.001) between the clusters.

A post-hoc clustering analysis was performed on primary serous carcinoma samples classified as grade 2 (*n* = 80) according to the classification of Silverberg [21]. Repeated *k*-means clustering resulted in two stable clusters: cluster C with high PI3K and low NF-κB pathway activity (*n* = 40) and cluster D with low PI3K and high NF-κB pathway activity (*n* = 40). There was no difference in PFS (*p* = 0.167) or OS (*p* = 0.077) between the clusters. Kaplan–Meier curves of the PFS and OS are provided in the Appendix A.

#### 2.2.2. GSE32062

To test cluster formation based on a combination of PI3K and NF-κB pathway activity, we subsequently analyzed the ten available HGSC samples of GSE32062. Repeated *k*-means clustering resulted in two stable clusters, containing five samples each. Cluster I had high PI3K and low NF-κB pathway activity, and cluster II had low PI3K and high NF-κB pathway activity. Survival curves of PFS (*p* = 0.144) and OS (*p* = 0.435) are plotted in Figure 3.

### 2.3. STA-Analysis of Molecular Subtypes as Defined by Tothill et al.

In their study, Tothill et al. identified six molecular subtypes of ovarian cancer based on expression profiling results of ovarian cancer samples of several histological subtypes [17]. In their analyses, the clusters were formed with *k*-means clustering and class prediction, based on Affymetrix microarray gene expression profiling, and were subsequently analyzed with gene function classification analyses to characterize the expression patterns of the newly formed clusters. As STA analysis is a novel approach, we decided to perform STA analysis on the Tothill clusters. Tothill et al. used the entire GSE9891 database for clustering, including borderline tumors and endometrioid carcinomas of the ovary. Thirty samples could not be allocated to a cluster, and thus were excluded from further analysis in their paper. To compare STA analysis results to the previously described molecular cluster characteristics formed by Tothill et al., we used all primary tumor samples allocated to one of the six clusters (*n* = 234). Boxplots of pathway activity per cluster for all pathways are provided in Figure 4.

Cluster 1 (*n* = 75) was termed the reactive stroma subtype by Tothill et al. It consisted almost exclusively of HGSC samples (99%) and was associated with the shortest PFS and OS. STA analysis showed that cluster 1 had the highest AR (*p* < 0.001; Figure 4a) and TGF-β (*p* < 0.001; Figure 4g) pathway activity compared to the other clusters.

Cluster 2 (*n* = 43) was marked by a high immune signature, and was composed mostly of HGSC samples, albeit at a lower percentage (93%) than cluster 1. When comparing STA results, cluster 2 was characterized by elevated NF-κB activity (*p* < 0.001; Figure 4e).

Cluster 3 (*n* = 28) was a low proliferative cluster that contained all borderline tumors, as well as ten serous carcinomas, three of which were low-grade serous carcinomas (LGSC; e.g., differentiation degree 1); in addition, four samples were grade 2, two samples were grade 3, and for one sample differentiation degree was unknown. Samples of this cluster were associated with a longer PFS and OS. In STA analysis, relatively high ER (*p* < 0.001; Figure 4b) and Wnt (*p* < 0.001; Figure 4h) pathway activity was seen in this cluster compared to the others. As cluster 3 is a combination of two radically different tumor types (i.e., borderline tumors and serous carcinomas), we examined STA-analysis results per tumor type. There were no differences in ER (*p* = 0.268) and Wnt (*p* = 0.654; Appendix A) pathway activation between the borderline tumors and serous carcinomas within the cluster.

Cluster 4 (*n* = 45) was characterized by a low stromal response, and samples were predominantly grade 2 or grade 3 serous carcinomas (89%). In STA analysis, this was the cluster associated with the lowest TGF-β pathway activity (*p* < 0.001; Figure 4g).

Cluster 5 (*n* = 35) was named the mesenchymal subtype and consisted predominantly of serous carcinomas of intermediate (40%) or high tumor grade (60%). Compared to other clusters, it had a lower AR (*p* < 0.001; Figure 4a) and ER (*p* < 0.001; Figure 4b) pathway activity, while HH pathway activity (*p* < 0.001; Figure 4d) was elevated. Furthermore, PI3K pathway activity was high compared to the other clusters (*p* < 0.001; Figure 4c). Even though Tothill et al. suggested possible Wnt pathway involvement, STA analysis showed no increase in Wnt pathway activity (*p* = 0.123; Figure 4h).

Cluster 6 (*n* = 8) contained mostly endometrioid carcinomas (88%) with low to intermediate differentiation degree. It was identified as the Wnt active cluster, and was associated with a longer PFS. Wnt pathway activity was the identifying trait (*p* < 0.001; Figure 4h).

## 3. Discussion

In this explorative study on signaling pathway activity in HGSC samples, we identified two subgroups with a difference in PFS and OS. The good-prognosis cluster was characterized by low PI3K and high NF-κB pathway activity, and the poor-prognosis cluster by high PI3K and low NF-κB pathway activity. Furthermore, we investigated signaling pathway activity in previously defined clusters by Tothill et al., in order to compare identifying traits and translate the cluster-specific gene profiles into clinically actionable signaling pathway activities [17].

### 3.1. PI3K and NF-κB Pathway Activity in HGSCs

In the newly-formed HGSC clusters, we found a more favorable prognosis in patients with lower PI3K and higher NF-κB pathway activity. Our finding of high PI3K pathway activity in the poor prognosis cluster is in line with PI3K pathway activation in a multitude of cancers, and the identification of its role as a “poor prognosis pathway” in cancers like non-small-cell lung cancer, colorectal cancer, and gastric cancer [22]. In ovarian cancer, increased PI3K activation, as measured by mammalian target of rapamycin (mTOR) phosphorylation levels, is seen in all stages of clear-cell carcinoma, and is considered to be an early event. In contrast, in serous carcinomas, increased mTOR phosphorylation levels were more commonly found in advanced stages of disease [23]. Given the role of PI3K in cell migration and invasion, pathway activation in serous carcinomas may have a more prominent role in tumor progression [24]. Additionally, PI3K pathway activation has been shown to be a prognostic marker for decreased survival [25,26].

For the association between high NF-κB pathway activity and improved survival, we suggest two possible underlying mechanisms. The first mechanism would be a difference in the balance between cell division (PI3K pathway activation) and apoptosis (NF-κB pathway activation). This would, however, be in contrast to previous studies on NF-κB in ovarian cancer, as it is often linked to a poorer survival [27]. However, NF-κB pathway activation in these studies is often based on the presence of pathway proteins, which may not be representative for actual pathway activity [7]. This is reflected in contradictory correlations between the different NF-κB pathway proteins and survival [28]. Furthermore, in low-grade serous ovarian carcinoma, NF-κB signaling seems to induce apoptosis [29]. If the increased survival of the NF-κB active cluster would be due to increased apoptosis, it is to be expected that the measured NF-κB pathway activity derives from cancer cells [30]. The second mechanism for the association between the increased NF-κB activity and improved survival could be result from a difference in immune response, as immune cells may have infiltrated the ovarian cancer tissue used to extract RNA for gene expression profiling. If this was the case, the elevated NF-κB activity would mainly result from immune infiltrate instead of actual cancer cells. NF-κB pathway activity could indicate a more effective immune response, which is associated with improved survival, as the presence of CD3+ tumor-infiltrating T cells has been associated with improved clinical outcome in women with advanced-stage epithelial ovarian cancer [31]. In the accompanying paper for GSE9891, Tothill et al. used immunohistochemistry to investigate intratumoral CD3+ cell infiltrate in tissue microarrays, revealing that the highest number of intratumoral CD3+ cells were within the samples of Tothill cluster 2 [17]. Nearly half of the samples in our high-NF-κB pathway activity cluster were classified as Tothill cluster 2, lending support to the immune hypothesis. This suggests that the measured NF-κB pathway activity is derived from immune infiltrate, which also explains why, in the paper of Tothill et al., cluster 2 had a relatively good prognosis. To determine whether NF-κB pathway activity is derived from cancer cells or immune infiltrate would require laser capture microdissection of cancer tissue samples, with separate pathway activity analysis for cancer cells and immune cells.

Clustering of the GSE32062 dataset also resulted in a cluster with low-PI3K and high-NF-κB pathway activity, with a more favorable prognosis, and a cluster with high-PI3K and low-NF-κB pathway activity with a less favorable prognosis, similar to the findings in the GSE9891 dataset. Although survival outcomes of the clusters in GSE32062 are very similar to the clusters in GSE9891, results did not reach statistical significance for this dataset, possibly as a result of the limited sample size (*n* = 10). As we had no information on tumor-infiltrating T cells, we were unable to substantiate any of our proposed underlying mechanisms.

In GSE9891, all samples were graded according to the classification system of Silverberg [17,21]. Although this classification system acknowledges three differentiation degrees for serous carcinomas, serous carcinomas are increasingly considered a dual entity, each with a different etiology and different response to treatment: ovarian cancer type I (LGSC and similar cancer types) and type II (HGSC and similar cancer types) [32]. In this division, grade 1 serous carcinomas are classified as LGSCm while grade 2 and grade 3 samples are classified as HGSC. In our analyses, clustering based on PI3K and NF-κB pathway activity only resulted in two clusters with a difference in PFS and OS if the inclusion was limited to grade 3 serous carcinomas. This is in line with the finding that some serous carcinomas classified as grade 2 more closely resemble LGSC, while they would be classified as HGSC based on tumor grade [33].

### 3.2. Comparing Molecular Subtypes as Formed by Tothill et al. to STA Analysis Results

We visualized pathway activity per cluster, demonstrating that our findings from STA analysis are in line with results as described by Tothill et al. [17]. For cluster 1, which had the lowest PFS and OS, Tothill et al. attributed the clustering primarily to stromal gene expression. They performed a more in-depth analysis of the samples allocated to cluster 1, which revealed a high level of moderate to extensive desmoplasia. This finding was reflected by elevated TGF-β pathway activity in STA-analysis, as the TGF-β pathway activates stromal tissue fibroblasts [34,35,36]. Cluster 2 was identified as a cluster with an elevated immune signature, corresponding to the NF-κB pathway activity in STA-analysis [30,37]. For cluster 3, the elevated ER pathway activity possibly reflects the larger proportion of low-grade tumors in this cluster (39%), as ER pathway activity is lost in higher tumor grades [38]. Additionally, Wnt pathway activity was relatively high compared to the other clusters, with exception of cluster 6. The combination of elevated ER and Wnt pathway activity was also seen in cluster 6, the second good-prognosis cluster. Although the Wnt pathway can act as a tumor-driving pathway, hyper-activation may promote apoptosis in cancer cells [39]. A study by Hoffmann et al. shows that an environment with low Wnt signaling is required for the development of HGSC organoids [40]. Thus, although Wnt has been described as tumor-driving, hyperactivation may result in a more favorable outcome for patients. Cluster 4 was associated with low stromal response by Tothill et al., which was consistent with our finding of low TGF-β pathway activity [34]. Cluster 5 was identified as a novel HGSC subtype with a mesenchymal expression pattern and an unfavorable prognosis in the paper by Tothill et al. The high HH and PI3K pathway activity in this cluster support cancer stem cell-like characteristics of tumors in this cluster [41,42,43]. In breast cancer, PI3K signaling is known to promote multipotency in cancer cells [44,45]. In addition, dedifferentiation of the cells is reflected by the low AR and ER pathway activity [46,47,48,49]. Although PANTHER pathway analysis, which is a comprehensive platform for gene function classification, suggests involvement of the Wnt pathway, we did not observe elevated Wnt pathway activity in STA-analysis [50]. One of the crucial differences between PANTHER and STA-analysis is that PANTHER is not designed to measure pathway activity in a single dataset, but rather to identify the differences between two or more groups of samples [10]. Furthermore, the result of PANTHER analysis is based on overrepresentation of a set of differentially expressed genes in a pathway list, while differences between mRNA, protein, and activated protein levels are not taken into account [10]. Thus, while these genes may be related to a specific signaling pathway, they do not necessarily reflect the functional activity of this pathway.

Cluster 6 was defined as a Wnt pathway-active cluster in the paper of Tothill et al., based on intense nuclear beta-catenin staining in these tumors and by PANTHER pathway analysis. As shown in Figure 4, STA-analysis confirmed this high-Wnt pathway activity in cluster 6 when compared to the others. Furthermore, ER pathway activity was elevated in a pattern similar to that of cluster 3.

### 3.3. Strengths and Limitations

In this study, we apply a new technique, STA analysis, on well-documented datasets with clinical annotations. The accompanying paper for GSE9891 by Tothill et al. provides ample additional information, such as the results of the reviewing of morphology and immunohistochemistry staining and results of tissue microarrays to count T cell infiltrate, allowing for a more in-depth interpretation of the STA analysis results. Furthermore, the additional follow-up data provided by the Australian Ovarian Cancer Study (AOCS) further confirmed the validity of the clusters with respect to survival. All included samples were taken prior to start of chemotherapy. As chemotherapeutic treatment affects STP activity, the results of our analyses may have been very different in samples taken after chemotherapy [51,52]. As we made use of publicly available databases, we were unable to revise pathology diagnoses. In the supplemental results of the paper for dataset GSE9891, Tothill et al. describe the pathology review of included samples. This was possible for 202 samples, while for 74 samples, pathology diagnosis including re-grading of samples, which according to the classification of Silverberg had to be done based on the pathology reports, it would have been preferable if all included samples had been reviewed by the same team of pathologists [17,21]. Prior to STA analysis, all samples were subject to an additional quality control. As all samples passed the control, they could all be included. However, for some comparisons, the number of included samples was small, urging for a more cautious interpretation of results. The main strength of STA analysis is the translation of gene expression profiling results into pathway activity scores that in principle are clinically actionable, meaning they can be targeted with drugs. For example, our newly formed poor-prognosis cluster with an active PI3K pathway could possibly be treated with PI3K pathway inhibitors.

### 3.4. Future Studies

STA analysis is currently being investigated in a multitude of other carcinomas and healthy tissues. We are working on the assembly of our own HGSC dataset, which allows us to define patient populations and collect even more data on treatment and recurrences. To establish a baseline of pathway activity in both the tissue of origin and the precursor lesion of HGSC, members of our group are working on STA analysis in healthy fallopian tube tissue and serous tubal intraepithelial carcinoma (STIC) lesions [53,54]. In addition, they are working towards the first use of STA analysis in a clinical setting, to determine aberrant pathway activity in HGSC patients with recurrent disease (NCT03458221). Eventually, we aim to identify aberrant pathways in the developing stages of HGSC, which in turn may be key in the development of targeted therapies, for the main value of STA analysis is the translation of cluster signatures from non-treatable gene patterns to clinically actionable pathways.

## 4. Materials and Methods

### 4.1. STA Analysis

The development and validation of the available test to measure activity of, respectively, the Wnt and ER pathways [7]; the AR, HH, TGF-β, and NF-κB pathways [11]; the Notch pathway [12,13]; and the PI3K–FOXO pathway [11,55] have been described in detail previously. In summary, the probability of a pathway-associated transcription factor to be actively transcribing its target genes is derived by a Bayesian computational network. This Bayesian network describes how measured probe set intensities depend on expression of target genes, which in turn depend on transcription complex activation [7]. STP activity is inferred from the activity of the transcription factor. For the creation of the computational pathway models, approximately 25 to 35 target genes per pathway were selected based on scientific literature [10].

For the PI3K pathway, pathway activity is inferred based on the FOXO transcription factor, as it is directly inverse to PI3K pathway activity in the absence of cellular oxidative stress [11]. In Affymetrix datasets, we used the expression level of the FOXO target gene *SOD2* to determine whether or not there was evidence for oxidative stress, and subsequently inferred the activity of the PI3K pathway, as previously described [11]. As there were only three samples with evidence of cellular oxidative stress, we decided to present PI3K values throughout this paper.

### 4.2. Datasets

We searched the Gene Expression Omnibus (GEO; https://www.ncbi.nlm.nih.gov/geo/) for publicly available datasets containing Affymetrix data of HGSCs with clinical annotations [16]. We limited our search to datasets containing Affymetrix HG-U133Plus2.0 data, as the pathway models have been built and calibrated on probesets on the array associated with the selected target genes. As we aim to relate pathway activity to clinical outcomes, individual patient data was required. Clinical annotations had to at least include PFS and OS, as well as information on when the sample was taken. Samples were eligible if they were taken prior to start of chemotherapy. Samples taken after start of chemotherapy were excluded, as it is known that the treatment affects STP activity in cancer cells [51,52]. We found two datasets suitable for analysis: GSSE9891, with an accompanying paper by Tothill et al. and GSE32062, with an accompanying paper by Yoshihara et al. All samples were subject to extensive quality control (QC) based on 12 quality parameters following Affymetrix recommendations and previously published literature [10,56,57]. All samples passed the QC, enabling use of the Affymetrix data as input for the pathway models to calculate the probability of pathways being either active or inactive. A normalized score with a range from 0 to 100 for pathway activity was provided for statistical analysis [14].

For GSE9891, we were able to obtain additional survival data from the Australian Ovarian Cancer Study (AOCS) (including data from the Westmead GynBiobank). The AOCS is an internationally available resource that collects biospecimens and extensive clinical outcome data on women enrolled in the study. As the study of Tothill et al. was published in 2008, we applied for additional follow-up data on women diagnosed with HGSC from this dataset. The AOCS study was approved by the Human Research Ethics Committees at the Peter MacCallum Cancer Centre (Human Research Ethics Project Number 16/161), The Westmead Institute for Medical Research, Queensland Institute of Medical Research, University of Melbourne and all participating hospitals. Our request for additional follow-up data as it was known in October 2019 was provided under approved AOCS Project #19/07.

### 4.3. Statistical Analysis

#### 4.3.1. Patient Characteristics

Frequencies of patient characteristics are presented as numbers and percentages. Normally distributed continuous variables are presented as means and standard deviations (SDs), and were analyzed with an independent sample *t*-tests. Skewed continuous variables are presented as medians and interquartile ranges (IQRs), and were analyzed with a Mann–Whitney U test.

#### 4.3.2. Clustering Analysis

We selected primary serous carcinoma samples classified as grade 3, according to the classification of Silverberg, and used repeated *k*-means clustering analyses based on pathway activity scores [21]. Survival curves were visualized in Kaplan–Meier curves, and a log-rank test was used to test for differences in survival between the newly formed clusters. After the provision of additional follow-up data by the AOCS, the survival analyses were repeated.

The classification of epithelial ovarian cancer has been subject to many changes over the years and has lacked a standardized grading system. This is further complicated by the more recently adopted division of serous carcinomas into either HGSCs or low-grade serous carcinomas (LGSCs), instead of grading serous carcinomas as grade 1, 2, or 3. Generally, grade 1 serous carcinomas are called LGSCs, while grade 2 and grade 3 serous carcinomas are called HGSCs. However, a proportion of grade 2 samples should be classified as LGSCs. In our clustering analysis, we have selected grade 3 carcinomas to lower heterogeneity. To investigate the effect of tumor grade on our results, we have additionally performed a post-hoc analysis of HGSC samples classified as grade 2 according to the classification of Silverberg [21].

#### 4.3.3. STA Analysis of Molecular Subtypes as Defined by Tothill et al.

Pathway activity per cluster is visualized with boxplots displaying the median and IQR, with overlaying scatterplots displaying the full range of individual samples. When comparing pathway activity among the Tothill clusters, we adopted a one-versus-the-rest approach, in which we compared each cluster to all other clusters combined. A Mann–Whitney U test was used to test for significance. A *p*-value of <0.05 was considered to be statistically significant. All statistical analyses were performed with SPSS (IBM Corp. Released 2016. IBM SPSS Statistics, version 25.0. IBM Corp., Armonk, NY, USA).

## 5. Conclusions

In this explorative study, we examined signaling pathway activity in ovarian cancer and identified two clusters based on NF-κB and PI3K activity, with a difference in PFS and OS in grade 3 HGSCs. The low-PI3K and high-NF-κB pathway activity of the favorable prognosis cluster may indicate a more active immune response, while the high-PI3K and low-NF-κB pathway activity of the unfavorable prognosis cluster may indicate high cell division. Elucidating underlying mechanisms of HGSC aides our understanding of the disease and provides possible new leads for targeted therapies.

## Figures and Tables

**Figure 1 cancers-12-02660-f001:**
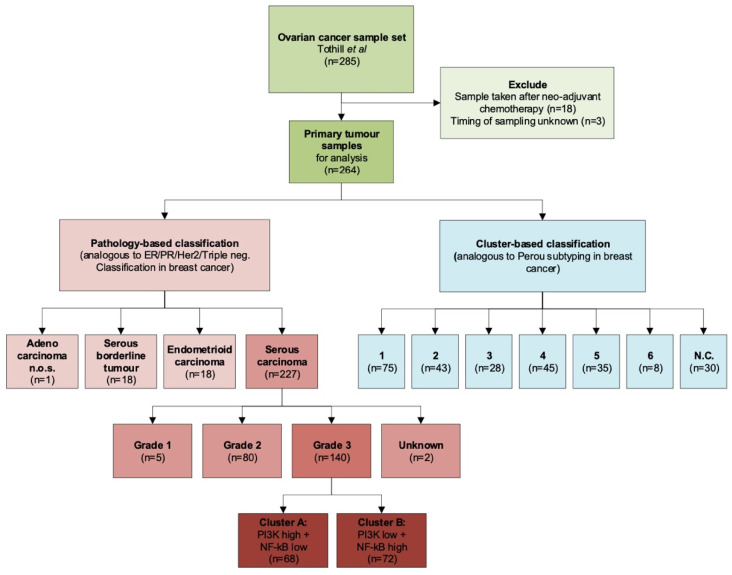
Overview of sample selection for GSE9891. Included samples are classified according to pathology diagnosis, or according to the clusters as formed by Tothill et al.

**Figure 2 cancers-12-02660-f002:**
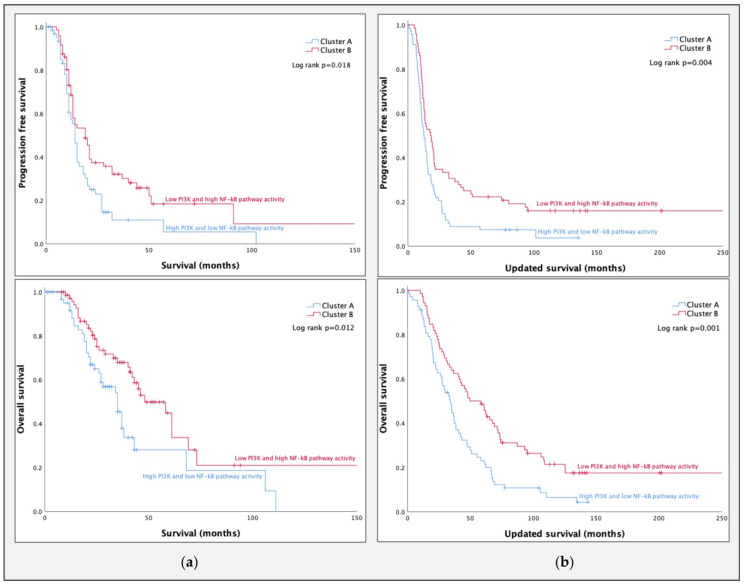
(**a**) Progression-free survival (PFS) and overall survival (OS) of the newly formed high-grade serous carcinoma (HGSC) clusters in dataset GSE98191 (*n* = 140) using the original follow-up data; (**b**) PFS and OS with additional follow-up data provided by the Australian Ovarian Cancer Study (AOCS).

**Figure 3 cancers-12-02660-f003:**
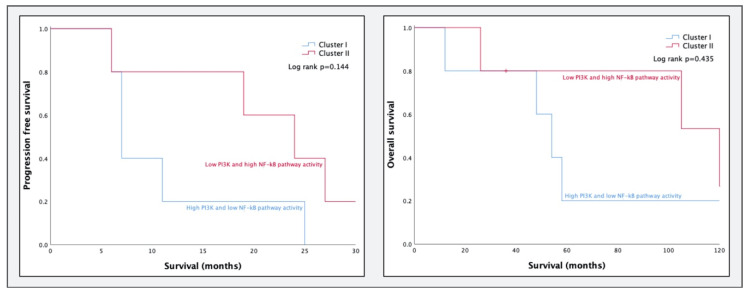
Progression-free survival and overall survival of the newly formed clusters in dataset GSE32062 (*n* = 10).

**Figure 4 cancers-12-02660-f004:**
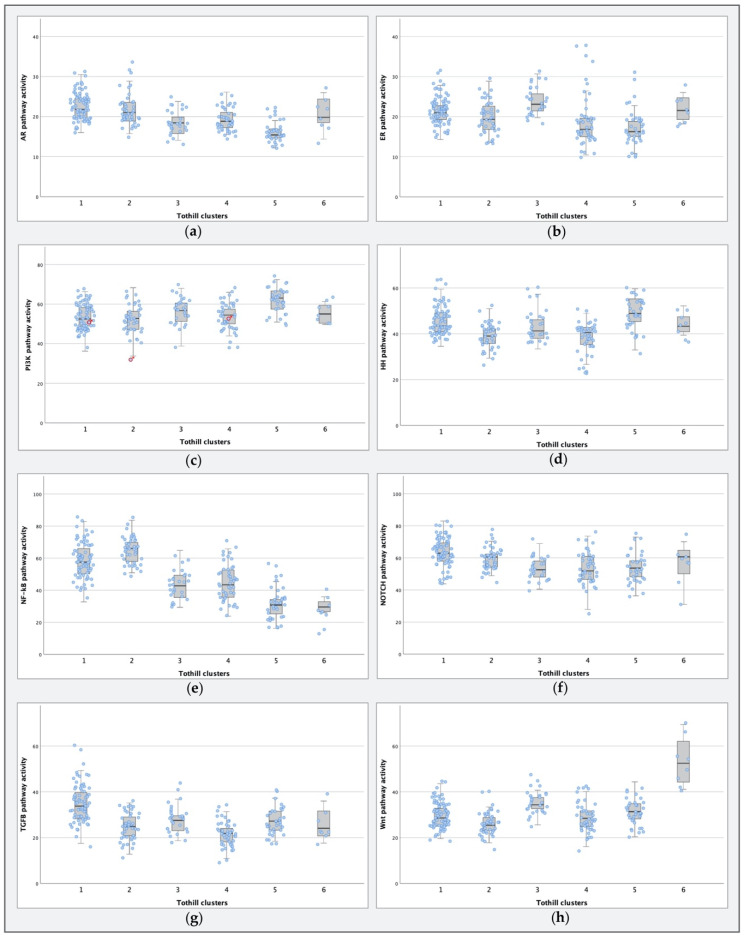
Boxplots for pathway activity per Tothill cluster, with overlying dot plots representing individual samples for the (**a**) androgen receptor (AR), (**b**) estrogen receptor (ER), and (**c**) phosphoinositide 3-kinase (PI3K) (the readout of samples marked with * (*n* = 3) should be interpreted with caution, due to evidence of cellular oxidative stress), (**d**) Hedgehog (HH), (**e**) nuclear factor-kappa B (NF-kB), (**f**) Notch, (**g**) transforming growth factor beta (TGF-β), and (**h**) Wnt pathways.

**Table 1 cancers-12-02660-t001:** Patient and tumor characteristics of primary tumor samples in dataset GSE9891.

Feature	Total	Tothill Clusters						
		1	2	3	4	5	6	NC^1^
Number of samples	264	75 (28%)	43 (16%)	28 (12%)	45 (17%)	35 (13%)	8 (3%)	30 (11%)
Age at diagnosis (y)^2^	60 (±11)	59 (±11)	62 (±10)	52 (±12)	60 (±9)	63 (±10)	54 (±6)	61 (±11)
Stage at diagnosis								
I	24 (9%)	1 (1%)	5 (12%)	10 (36%)	1 (2%)	2 (6%)	5 (62%)	
II	18 (7%)	1 (1%)	4 (9%)	4 (14%)	5 (11%)	2 (6%)	1 (13%)	1 (3%)
III	206 (78%)	65 (87%)	31 (72%)	13 (46%)	39 (87%)	30 (85%)	2 (25%)	26 (87%)
IV	16 (6%)	8 (11%)	3 (7%)	1 (4%)		1 (3%)		3 (10%)
Histology								
Serous carcinoma	227 (86%)	74 (99%)	40 (93%)	10 (36%)	40 (89%)	34 (97%)	1 (12%)	28 (93%)
Endometrioid carcinoma	18 (7%)	1 (1%)	2 (5%)		5 (11%)	1 (3%)	7 (88%)	2 (7%)
Serous borderline tumor	18 (7%)			18 (64%)				
Adenocarcinoma NOS^3^	1 (0%)		1 (2%)					
Differentiation degree								
1	17 (7%)	1 (1%)	2 (5%)	11 (39%)			3 (38%)	
2	88 (33%)	25 (34%)	13 (30%)	4 (15%)	19 (42%)	14 (40%)	4 (50%)	9 (30%)
3	156 (59%)	48 (64%)	28 (65%)	11 (39%)	26 (58%)	21 (60%)	1 (12%)	21 (70%)
Unknown	3 (1%)	1 (1%)		2 (7%)				
Status								
Deceased	103 (39%)	44 (59%)	14 (33%)	1 (4%)	16 (36%)	15 (43%)		13 (43%)
Deceased, other cause	2 (1%)	1 (1%)			1 (2%)			
Progressed	70 (26%)	19 (25%)	17 (39%)	3 (11%)	14 (31%)	9 (26%)		8 (27%)
Progression-free	89 (34%)	11 (15%)	12 (28%)	24 (85%)	14 (31%)	11 (31%)	8 (100%)	9 (30%)

All data are presented as mean and standard deviation or as frequency and percentage. ^1^ Not classified; ^2^ years; ^3^ not otherwise specified.

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
