# Peer review of "Signal Transduction Pathway Activity in High-Grade, Serous Ovarian Carcinoma Reveals a More Favorable Prognosis in Tumors with Low PI3K and High NF-κB Pathway Activity: A Novel Approach to a Long-Standing Enigma"

_cancers, 2020, doi:10.3390/cancers12092660_

Round 1
Reviewer 1 Report
In this manuscript the authors have performed a STA analysis on two ovarian cancer datasets. They further correlated the STA outcome with OS and PFS in ovarian caner.
Authors have presented that data very clearly and logically. It is a straightforward analysis of the publicly available dataset for ovarian cancer.
However, there are some major concerns in this manuscript.
- It is not clear from the manuscript that what were the search criteria/keywords used for searching the GEO database? How many studies/datasets were initially identified based on the search criteria and which filters/criteria used that resulted in the two particular studies used in this manuscript?
- The main focus and discussion point in this manuscript is from Tothhill et al. For a significant part of the manuscript authors refer back to the original article of Tothill et al published in 2008, therefore it is unclear what exactly is the originality and value addition from the analysis in this manuscript?
- Tothill et al was published in 2008, did authors not find any other datasets since 2008 which could be analyzed? Inpast decade there has a large number of studies, including efforts from TCGA, which provides a comprehensive gene expression datasets and made publicly available. Did authors not find TCGA datasets useful for their study and if so than what were the basis of its exclusion? In Australia there are groups which have made significant efforts to curate datasets for ovarian cancer, did authors look into those?
- There is a clinical trial (ClinicalTrials.gov Identifier: NCT03458221) for STA in ovarian cancer. Did authors know about it? I would be good to discuss this trial in the manuscript.
- Out of the only two datasets used in this manuscript, results from Yoshihara are insignificant (due to low sample number n=10) so the manuscript is based on only one dataset from 2008. It is recommended that authors provide justification why the 2008 study is the only study that could be used for their analysis and no other study till 2020 isn't useful for this analysis? And if the 2008 dataset is the only useful dataset for the STA analysis then authors should justify that how would their study be of relevance to many datasets and studies published on ovarian cancer in the last decade.
- Did authors try to use this approach on ICLE dataset? can it be applied to the cell line expression data?
Minor:
Line 122: Is Section 2.2 missing or is it incorrect numbering?
Author Response
General comments:
In this manuscript the authors have performed a STA analysis on two ovarian cancer datasets. They further correlated the STA outcome with OS and PFS in ovarian cancer. Authors have presented that data very clearly and logically. It is a straightforward analysis of the publicly available dataset for ovarian cancer. However, there are some major concerns in this manuscript.
Response to general comments:
Thank you very much for your time and effort in the reviewing of our manuscript. We are happy to hear that we were able to present our date in a clear and logical manner and are grateful for the opportunity to address your concerns and improve our manuscript accordingly. In the text below, we have provided a point-by-point response.
Point-by-point response:
Major concerns:
- It is not clear from the manuscript that what were the search criteria/keywords used for searching the GEO database? How many studies/datasets were initially identified based on the search criteria and which filters/criteria used that resulted in the two particular studies used in this manuscript?
For the selection of datasets, we had two important criteria to consider. The first is that the signaling pathway models (assays) were developed using Affymetrix HG-U133Plus2.0 data, and thus are suitable for data generated with this platform. Although it would be possible to adapt the pathway assays to other microarray platforms, they currently are not available nor validated. Our second criterium was the availability of individual patient data as we aimed to correlate pathway activity scores to clinical outcomes. When searching the GEO database for datasets containing Affymetrix HG-U133Plus2.0 data (Accession GPL570) and use the keyword ‘ovarian’ there are 166 results, but to the best of our knowledge, only 2 sets have a supplementary file containing individual patient data. Thus although, there were other ovarian cancer datasets of Affymetrix HG-U133Plus2.0 data, these provided insufficient clinical data or no individual patient data to answer our research question and were not suitable for analysis. As we have provided insufficient clarification in our manuscript, we elaborated on the criteria in the methods section (lines 362 to 365).
- The main focus and discussion point in this manuscript is from Tothill et al. For a significant part of the manuscript authors refer back to the original article of Tothill et al published in 2008, therefore it is unclear what exactly is the originality and value addition from the analysis in this manuscript?
The additional value of our analysis is that we assess the data with a new technique which translates gene expression profiling results into clinically actionable signal transduction pathway activity scores instead of discovering subsets of patients characterized by a common gene expression pattern. The latter approach is not clinically actionable, generally hard to reproduce and is not suited to perform on a single patient sample. The here used pathway activity assays have been biologically validated before and, as they can be performed on a single sample, can be used as a diagnostic method when a patient is diagnosed with HGSC.
Initially, we intended to only present the results of our own clustering analysis based on signaling pathway activity. However, we felt that STA analysis of the previously formed Tothill clusters provided interesting insights as we were able to complement some of their findings (such as the ‘wnt active’ gene expression cluster, which indeed appeared to be characterized by an active Wnt pathway). To avoid confusion on when we are discussing our own results and when we are comparing them to the results of Tothill et al, we replaced the dataset names with their GEO accession numbers throughout the manuscript. We now only mention the ‘Tothill clusters’ when comparing our STA results to the clusters that were formed in the original manuscript of Tothill et al.
- Tothill et al was published in 2008, did authors not find any other datasets since 2008 which could be analyzed? In past decade there has a large number of studies, including efforts from TCGA, which provides comprehensive gene expression datasets and made publicly available. Did authors not find TCGA datasets useful for their study and if so than what were the basis of its exclusion? In Australia there are groups which have made significant efforts to curate datasets for ovarian cancer, did authors look into those?
The pathway models are calibrated to Affymetrix HG-U133Plus2.0 data and thus cannot be applied to data from other platforms (yet), as discussed in response to point 1. Currently the pathway assays are being adapted to use on RNAseq data, we are looking forward to this development as it will enable analysis of TCGA data in the future. As for the Australian Ovarian Cancer Study Group, we have involved them in our research as they have provided additional follow-up data (up to 2020) for samples of GSE9891.
- There is a clinical trial (ClinicalTrials.gov Identifier: NCT03458221) for STA in ovarian cancer. Did authors know about it? I would be good to discuss this trial in the manuscript.
We are aware of this trial as it was initiated by part of our research group, we have included a brief discussion on the trial in the discussion section of the manuscript in line 338 to 340.
- Out of the only two datasets used in this manuscript, results from Yoshihara are insignificant (due to low sample number n=10) so the manuscript is based on only one dataset from 2008. It is recommended that authors provide justification why the 2008 study is the only study that could be used for their analysis and no other study till 2020 isn't useful for this analysis? And if the 2008 dataset is the only useful dataset for the STA analysis then authors should justify that how would their study be of relevance to many datasets and studies published on ovarian cancer in the last decade.
We would also like to refer here to the answer to the first comment. STA-analysis is unique in the way that it can determine pathway activity within a single sample and has been biologically validated for use on multiple cell and tissue types. Its intended use is not appliance to datasets but rather to single samples taken from patients at the time of diagnosis. This public dataset analysis is a first step into understanding pathway activity in HGSC. Eventually we hope to identify aberrant pathways within individual patients, to provide targets for individualized therapies. We are currently working on such a trial as mentioned under point 4.
- Did authors try to use this approach on ICLE dataset? can it be applied to the cell line expression data?
The pathway models can be applied to cell line expression data. However, in this case we aimed to relate pathway models to clinical outcome data and characterize HGSC in terms of combined signaling pathway activity for the first time. As a result, cell line expression data was not suitable for inclusion in our analysis. However, in the references with regard to development of the pathway assays there are many examples of application to cell lines, which is well possible.
Minor concerns:
- Line 122: Is Section 2.2 missing or is it incorrect numbering?
Apologies, the numbering is incorrect. We’ve made sure to adjust the numbering.
Reviewer 2 Report
The
The study by Lieshout, et al. describes their use of Signal Transduction Pathway Analysis (STA-analysis) against two published gene expression datasets to better clarify pathway activity that drives tumor progression and leads to differences in progression-free and overall survival.
Major comments:
1) The idea that NF-kB enhances survival due to immune infiltration could be supported by providing the immunohistochemical analysis provided in the Tothill paper, rather than requiring the reader to refer back to that paper. Additionally, the authors should provide an additional bioinformatics analysis of published immune cell gene signatures to deduce whether specific immune cell subsets with known NF-kB activity are present in the high NF-kB gene set. This is critical as it has been shown by multiple groups that NF-kB activation supports ovarian cancer progression.
2) Given the lack of functional data provided to support the bioinformatics analysis, the manuscript could be strengthened by including a discussion of published papers with mechanistic data that support the finding that high NF-kB and low PI3K are beneficial for survival in ovarian cancer models.
Minor comments:
1) The methods section states that samples were only eligible if they were taken prior to the start of chemotherapy, however it is unclear what treatments were given to the patients subsequently for the Tothill set. The Yoshihara set is described as all receiving debulking surgery and treatment with chemotherapy. Is this also the case with the Tothill set? These are important parameters that could affect OS and PFS.
2) Although this study focuses on pathways that are active prior to chemotherapy and acknowledges that treatment affects STP activity in cancer cells, this is not discussed as a limitation of the study. Given the strong influence of the tumor microenvironment on signal transduction pathways and immune cell infiltration, it would be worth noting how different clusters may emerge in the post-chemotherapy setting.
3) Was the reason for focusing on grade 3 for clustering analysis of the Tothill set? The methods section describing the STA-analysis does not sufficiently describe how “high” and “low” pathway activity is delineated? Were there no samples in this dataset where both pathways (PI3K or NF-kB) were high or low?
Author Response
General comments:
The study by Lieshout, et al. describes their use of Signal Transduction Pathway Analysis (STA-analysis) against two published gene expression datasets to better clarify pathway activity that drives tumor progression and leads to differences in progression-free and overall survival.
Response to general comments:
Thank you for your time and effort in the reviewing of our work. We appreciate the importance of the raised concerns and are grateful for the opportunity to address them in our manuscript, which has benefitted substantially. In the text below, we have provided a point-by-point response to your comments including the lines of the manuscript where the changes were made.
Point-by-point response:
Major concerns:
- The idea that NF-kB enhances survival due to immune infiltration could be supported by providing the immunohistochemical analysis provided in the Tothill paper, rather than requiring the reader to refer back to that paper. Additionally, the authors should provide an additional bioinformatics analysis of published immune cell gene signatures to deduce whether specific immune cell subsets with known NF-kB activity are present in the high NF-kB gene set. This is critical as it has been shown by multiple groups that NF-kB activation supports ovarian cancer progression.
For the NF-kB activation, we are aware of the association between NF-kB gene expression and a poorer prognosis in ovarian cancer, however contradictory correlations between NF-kB pathway proteins and survival have also been reported (lines 238 to 239). In case we measured NF-kB activity in in cancer cells, it could also suggest that increased gene expression or presence of pathway proteins may not equal actual pathway activity. Furthermore, the GSE9891 dataset contained samples consisting of a mixture of cancer tissue cell types. Thus, the identified NF-kB pathway activity could also be derived from immune cells. In this case, NF-kB pathway activity could indicate a more effective immune response, which is associated with improved survival: the presence of CD3+ tumor-infiltrating T cells has been associated with improved clinical outcome in women with advanced stage epithelial ovarian cancer. As per your suggestion, we went back to the accompanying paper for GSE9891, in which Tothill et al. used immunohistochemistry to investigate intratumoural CD3+ cell infiltrate in tissue microarrays. They revealed that the highest number of intratumoural CD3+ cells were present within Tothill cluster 2, confirming that cluster 2 was the subtype with highest tumor infiltrate. Nearly half of the samples in our high NF-kB pathway activity cluster were classified as Tothill cluster 2. This lends support to our proposed immune hypothesis, that is, that the NF-kB pathway activity, at least in this cluster is mostly derived from the infiltrate. It also explains why this subtype of the Tothill set had a relatively good prognosis. In the manuscript, we have elaborated on the role of NF-kB and cancer in lines 234 to 240 and on the analysis by Tothill et al and its implications for our results in lines 248 to 256.
- Given the lack of functional data provided to support the bioinformatics analysis, the manuscript could be strengthened by including a discussion of published papers with mechanistic data that support the finding that high NF-kB and low PI3K are beneficial for survival in ovarian cancer models.
In the newly formed HGSC clusters, we found a more favorable prognosis in patients with lower PI3K and higher NF-kB pathway activity. The high PI3K pathway activity in the poor prognosis cluster is in line with PI3K pathway activation in a multitude of cancers and the identification of its role as a ‘poor prognosis pathway’ in cancers such as non-small cell lung cancer, colorectal cancer and gastric cancer. In ovarian cancer, increased PI3K pathway activation, as measured by mTOR phosphorylation levels, is seen in all stages of clear cell carcinoma and considered to be an early event. In contrast, in serous carcinomas increased mTOR phosphorylation levels were more commonly found in advanced stages of disease, indicative of a poor prognostic role. Given the role of PI3K in cell migration and invasion, pathway activation in serous carcinomas may have a prominent role in tumor progression. Additionally, PI3K pathway activation has been shown to be a prognostic marker for a decreased survival. In lines 225 to 231, we have added a more in-depth discussion on the role of PI3K. The NF-kB pathway activity was probably mostly associated with the presence of immune infiltrate, as discussed under point 1. The combination of low PI3K pathway activity with active immune infiltrate is a logical combination for a better prognosis.
Minor concerns:
- The methods section states that samples were only eligible if they were taken prior to the start of chemotherapy, however it is unclear what treatments were given to the patients subsequently for the Tothill set. The Yoshihara set is described as all receiving debulking surgery and treatment with chemotherapy. Is this also the case with the Tothill set? These are important parameters that could affect OS and PFS.
Almost all patients included in the Tothill dataset were treated with debulking surgery and chemotherapy. We should have included this in the manuscript as it indeed is very important in the survival of ovarian cancer patients. We have added this to the results section in lines 102 to 104. - Although this study focuses on pathways that are active prior to chemotherapy and acknowledges that treatment affects STP activity in cancer cells, this is not discussed as a limitation of the study. Given the strong influence of the tumor microenvironment on signal transduction pathways and immune cell infiltration, it would be worth noting how different clusters may emerge in the post-chemotherapy setting.
In our study we only included samples taken prior to start of chemotherapy as we aimed to related pre-treatment STP activity to clinical outcomes, which is why initially we did not mention it as a limitation of the study. However, it does limit the generalizability of the results and should definitely be discussed. We added it to the limitations section in lines 316 to 319. It would be very interesting to assess the effect of chemotherapy on STP activity and perhaps even correlate survival to change in STP activity from pre to post chemotherapeutic treatment in a follow-up study. - Was the reason for focusing on grade 3 for clustering analysis of the Tothill set? The methods section describing the STA-analysis does not sufficiently describe how “high” and “low” pathway activity is delineated? Were there no samples in this dataset where both pathways (PI3K or NF-kB) were high or low?
Ovarian cancer is notorious for its heterogeneity and is affected by many external factors. The pathological classification of ovarian cancer still lacks standardization and several systems have been in use that are not interchangeable. Although serous carcinomas are increasingly classified as LGSC and HGSC (which are radically different in clinical behavior), the existence of a former ‘middle’ group indicates that a group of tumors may not be allocated so easily. Indeed, while grade 2 and 3 carcinomas (graded according to the classification of Silverberg) generally are considered HGSC, recent studies show that a number of these should actually be classified as LGSC. To lower heterogeneity within our sample set we decided to select grade 3 carcinomas. We added our considerations with regards to this issue in lines 398 to 407.
As for the pathway activity, our description of ‘high’ and ‘low’ was derived from the K-means clustering results. As this is one of the first studies using STA-analysis on ovarian cancer it is unclear which pathway activity scores are normal for this tissue type and what is to be considered high or low. We have clarified this in lines 135 to 139.
Round 2
Reviewer 1 Report
Authors' response to the queries is satisfactory. The responses to the queries and the edits in the manuscript have provided further clarity to the content of the manuscript. There are no further comments. Thanks.